# Creation of flexible spin-caloritronic material with giant transverse thermoelectric conversion by nanostructure engineering

Ravi Gautam[1], Takamasa Hirai [1], Abdulkareem Alasli [2], Hosei Nagano[2], Tadakatsu Ohkubo [1], Ken-ichi Uchida [1]✉ & Hossein Sepehri-Amin [1]✉

Functional materials such as magnetic, thermoelectric, and battery materials have been revolutionized through nanostructure engineering. However, spin caloritronics, an advancing field based on spintronics and thermoelectrics with fundamental physics studies, has focused only on uniform materials without complex microstructures. Here, we show how nanostructure engineering enables transforming simple magnetic alloys into spin-caloritronic materials displaying significantly large transverse thermoelectric conversion properties. The anomalous Nernst effect, a promising transverse thermoelectric phenomenon for energy harvesting and heat sensing, has been challenging to utilize due to the scarcity of materials with large anomalous Nernst coefficients. We demonstrate a remarkable ~70% improvement in the anomalous Nernst coefficients (reaching ~3.7 μVK⁻¹) and a significant ~200% enhancement in the power factor (reaching ~7.7 μWm⁻¹K⁻²) in flexible Fe-based amorphous materials by nanostructure engineering without changing their composition. This surpasses all reported amorphous alloys and is comparable to single crystals showing large anomalous Nernst effect. The enhancement is attributed to Cu nano-clustering, facilitating efficient transverse thermoelectric conversion. This discovery advances the materials science of spin caloritronics, opening new avenues for designing high-performance transverse thermoelectric devices for practical applications.

Thermoelectric devices have emerged as a promising technology for generating electricity by scavenging waste heat, representing a vital step towards achieving a more sustainable future[1–3]. These devices leverage thermoelectric effects, enabling the direct interconversion between heat and electricity. Additionally, heat currents can also interact with spin-dependent transport, giving rise to the field of spin-caloritronics, which focuses on the interconversion of spin, charge, and heat currents[3–5]. However, current technology in this realm relies on a complex module structure based on the Seebeck effect, which has inadequate thermal insulation[4,6]. To overcome this problem, research interest in transverse thermoelectric conversion such as the

anomalous Nernst effect (ANE) has been raised[3,5]. ANE can convert a temperature gradient into an orthogonal voltage in magnetic materials (Fig. 1) and is the Onsager reciprocal of the anomalous Ettingshausen effect (AEE)[3,4,7–12]. ANE is one of the hottest topics in spin caloritronics[4,5,13] because it enables the design of thermoelectric devices with a simple lateral structure, convenient scalability, and easy fabrication. Additionally, its transverse geometry facilitates efficient and flexible coverage of curved heat sources, making it ideal for harvesting thermal energy from large-area heat sources. Compared to conventional Seebeck-effect-based devices, ANE-based devices require fewer manufacturing processes, exhibit lower contact

[1]National Institute for Materials Science, Tsukuba 305-0047, Japan. [2]Department of Mechanical Systems Engineering, Nagoya University, Nagoya 464-8601, Japan. ✉e-mail: uchida.kenichi@nims.go.jp; h.sepehriamin@nims.go.jp

**Fig. 1 | Anomalous Nernst effect in a nanostructure-engineered magnetic material. a–d** Schematic illustrations of the anomalous Nernst effect (ANE). $\nabla T$, **E**, and **M** denote temperature gradient applied to the sample, the electric field driven by ANE, and a unit vector of the sample magnetization. It demonstrates nanostructure-engineering of ANE in four different scenarios: (**a**) Fe-based amorphous sample (Nanomet), (**b**) the emergence of nanoprecipitates/clusters in an amorphous matrix, (**c**) the high-volume density of nanoprecipitates/clusters embedded in an amorphous matrix, indicating high output signals, and (**d**) the crystallized Nanomet samples with reduced ANE signals. The same scenario is applicable to the anomalous Ettingshausen effect (AEE) since ANE is the Onsager reciprocal of AEE.

resistance, and may offer superior thermoelectric conversion performance[2,14]. This makes ANE a promising path for the development of next-generation thermoelectric devices with potential applications ranging from thermal management technologies to heat flux sensors[2]. However, further improvement of the thermopower of ANE is needed, which is the current bottleneck for its practical applications. As the demand for modern devices continues to rise, achieving a significant transverse thermoelectric response and understanding the underlying mechanisms become crucial for optimizing efficiency and power generation capabilities. To accomplish this, a novel approach is required to design low-cost and flexible materials that exhibit large ANE, thereby advancing its use in cutting-edge thermoelectric devices.

The discovery of remarkable ANE in magnetic topological materials has triggered a surge of interest, and significant efforts have been directed toward enhancing ANE through materials research. For example, several magnetic materials such as Heusler compounds[7–9,15–19], ferromagnetic binary alloys[10,12,14,20–28], Weyl antiferromagnet[29–31], and permanent magnets[11,32–34] have been extensively investigated. However, only a few of these materials have displayed the anomalous Nernst coefficient $S_{ANE}$, transverse thermopower due to ANE, surpassing 1 $\mu VK^{-1}$, with Co-based Heusler alloys demonstrating particularly remarkable values of 6–8 $\mu VK^{-1}$ at room temperature[8]. Additionally, while most research has predominantly focused on crystalline materials, there has been a recent emergence of studies exploring amorphous materials[34–40]. Iron ($\alpha$-Fe) is known as the most abundant ferromagnetic element, but it exhibits tiny ANE[10,12,22]. Thus, it would be favorable to develop a versatile Fe-based bulk material with large ANE to obtain highly efficient thermoelectric devices. So far, particular emphasis has been directed on material design and development to increase the ANE via manipulating the Berry curvature near the Fermi level[11,17,19,28,29]. However, little attention has been paid to how microstructure engineering at different length scales can influence the transport properties and ANE. This can potentially unveil new strategies to enhance ANE, thus advancing our understanding and application of this effect in thermoelectric and energy conversion devices.

In this study, we demonstrate how nanostructure engineering of magnetic materials can significantly improve the transverse thermoelectric conversion properties. We present a schematic representation of our strategies in Fig. 1. To confirm the validity of our approach, we demonstrated this concept by investigating ANE in Fe-based amorphous materials, commercially known as "Nanomet"[41], by tailoring their nanostructure. Nanomet is a renowned soft magnetic material that contains a high amount of Fe (~ 83-85 at%) and cost-effective alloying elements (i.e., Cu, B, P, and Si). This material is widely employed in high-frequency power electronic devices, such as inductors and transformers, owing to its excellent soft magnetic properties. Surprisingly, we found that, although the as-prepared Fe-based amorphous alloy initially exhibited low ANE, optimal annealing dramatically increased $S_{ANE}$ despite the same average composition, owing to the formation of Cu nanoclusters with a large volume fraction. Notably, this material exhibits largest $S_{ANE}$ among amorphous alloys[34–40]. The study involved four steps: (a) starting with the as-quenched Nanomet samples with small ANE, (b) forming Cu clusters in the amorphous samples, (c) optimizing the density of Cu-rich clusters embedded in an amorphous matrix with a significant enhancement of ANE, and (d) transforming the samples into crystallized Nanomet ribbons with small ANE with further increasing annealing temperature. These strategies were devised to understand the influence of nanostructure on ANE/AEE and provide insights into the thermoelectric properties of the studied materials. Our work demonstrates that nanosized Cu-rich clusters in an amorphous matrix of Fe-based soft magnetic materials (Nanomet) lead to a significant 70% enhancement of $S_{ANE}$ without altering the material chemistry. This strategy resulted in the figure of merit for ANE comparable to the record-high value in a Heusler ferromagnet at room temperature[7]. The developed materials are easily mass-produced at a low cost, and their mechanical flexibility enables the fabrication of large-area ANE and AEE devices with various

shapes. This study and proof of concept thus open new avenues for enhancing the ANE/AEE properties of materials in general through a nanostructure engineering approach and discusses the correlation between the nanostructure and its transport properties.

## Results

### Nanostructure engineering of amorphous Fe-based ribbons

We have fabricated $18 \pm 2\,\mu m$ thin and several meters long Nanomet ribbons using the melt-spinning technique which brings great benefits for low manufacturing and raw material costs. Differential scanning calorimetry analysis of the Nanomet ribbons showed the crystallization of $\alpha$-Fe(Si) and Fe-compound phases at respective onset temperatures of $T_{x1} = 661\,K$ and $T_{x2} = 806\,K$ (Supplementary Fig. 1). Phase analysis of the as-quenched (as-spun) sample by X-ray diffraction (XRD) exhibits a broad halo peak indicating the presence of a fully amorphous phase (Supplementary Fig. 2) which was further confirmed by the electron diffraction pattern obtained using transmission electron microscopy (TEM). Figure 2a displays a high-resolution bright field (BF) scanning (S)TEM image obtained from the as-quenched sample. The uniform supersaturated solid solution with an amorphous structure was observed in this sample. The amorphous ribbons were annealed at various isothermal temperatures (573, 623, 643, 653, 673, and 723 K) for only 3 min to achieve different microstructural features and investigate their correlations with ANE. The microstructure of the annealed samples was studied by high-resolution BF-STEM and electron beam diffraction analysis as shown in Fig. 2b–e. Annealing at temperatures below 653 K does not change the amorphous microstructure, while annealing at temperatures above 673 K leads to the crystallization of the $\alpha$-Fe(Si) phase (Supplementary Fig. 2 and Fig. 3). The elements P, B, and C present in the Nanomet alloy composition, are known to be amorphous forming elements[42] and this is the main reason for the formation of an amorphous phase in the as-quenched ribbons, which inhibits the nucleation of $\alpha$-Fe(Si). Note that the obtained amorphous structure provides excellent flexibility to the ribbons developed. As the temperature was increased beyond the

crystallization points, there was a notable reduction in the presence of amorphous phases, accompanied by an increase in the nucleation of the $\alpha$-Fe(Si) phase. Subsequent annealing at 723 K resulted in a significant volume fraction of the $\alpha$-Fe(Si) phase with a larger grain size. Detailed examination through high-resolution TEM images revealed the distinct separation of the $\alpha$-Fe(Si) crystals from each other by the residual amorphous matrix. Moreover, the XRD analysis provided evidence for the appearance of additional diffraction peaks corresponding to the crystallization of the Fe-P and Fe-B compounds.

### Nano-scale elemental analysis of Fe-based ribbons

A question is if there is any nano-scale elemental fluctuation in an amorphous matrix before and/or after crystallization of $\alpha$-Fe(Si) crystals. To answer this question, we conducted atom probe tomography (APT). 3D elemental maps of Fe, P, B, and Cu are shown in Fig. 3a–d. The illustrated maps are ~10 nm thin sliced from whole data for a better visualization of the elemental distributions. A uniform distribution of all elements was observed in the as-quenched sample as shown in Fig. 3a, indicating its chemically homogeneous solid solution. As shown in Fig. 3b, a heterogeneity in the distribution of Cu atoms in the microstructure of the annealed sample at 623 K was detected. This observation strongly implies the occurrence of Cu clustering, indicated by the segregation of Cu atoms. As the annealing temperature was raised to 653 K, Cu clustering became more visible as demonstrated in Fig. 3c, with an increase in both the size and number density of nano-sized Cu-rich clusters (Supplementary Fig. 4). The average size of the Cu clusters for the Nanomet sample annealed at 653 K was determined to be approximately 2.2 nm, with an estimated number density of $7.3 \times 10^{23}\,m^{-3}$. Notably, no crystallization was observed up to 653 K (Fig. 2d), indicating the presence of Cu clusters dispersed within the amorphous matrix. However, due to the absence of discernible fringe contrast in the nanobeam TEM analysis, the specific structure of these clusters could not be identified. The clustering phenomenon of Cu in Fe-based material can be elucidated by considering the markedly positive enthalpy of mixing exhibited by this system. The low solubility

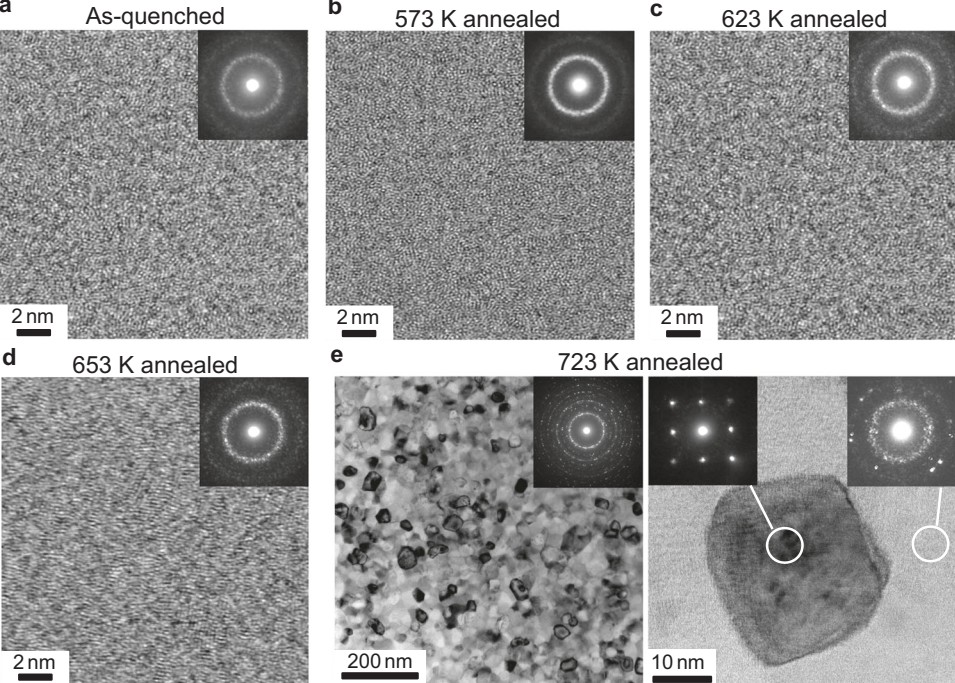

**Fig. 2 | High resolution TEM analysis of annealed Nanomet samples.** High-resolution bright field (BF)-STEM images with corresponding nanobeam electron diffraction patterns obtained from **a**, as-quenched ribbons and post-annealed at temperatures of 573, 623, 653, and 723 K shown in **b**–**e**, respectively. BF-STEM images of the 723 K annealed sample evidence the crystallization of $\alpha$-Fe(Si) along with residual amorphous regions.

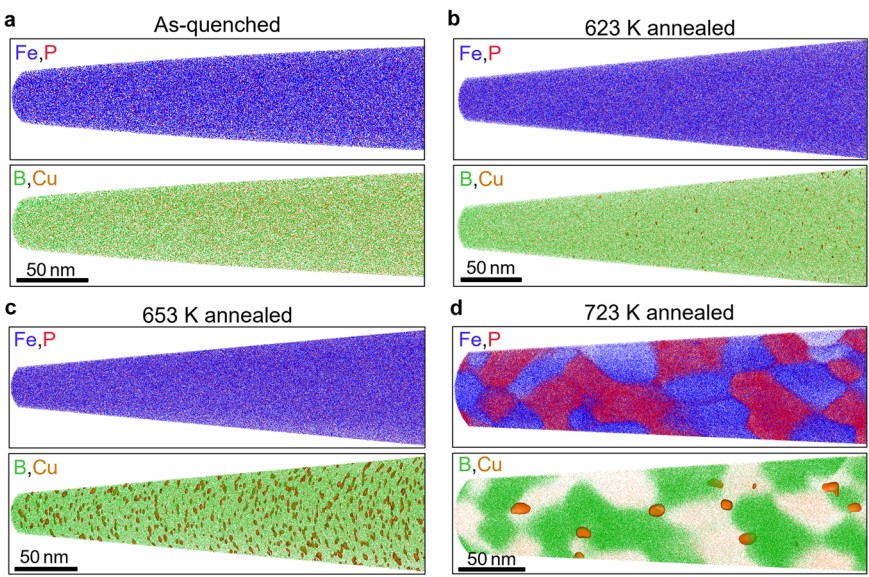

**Fig. 3 | Atom probe tomography analysis of annealed Nanomet samples.** APT elemental maps of Fe (blue), P (red), B (green), and Cu (orange) in different samples: (**a**) as-quenched, (**b**) annealed at 623 K, (**c**) annealed at 653 K, and (**d**) annealed at 723 K.

of Cu in Fe leads to the formation of Cu-enriched clusters within the amorphous phase, preceding the nucleation of primary $\alpha$-Fe crystals. These clusters then act as heterogeneous nucleation sites for the primary $\alpha$-Fe(Si) crystals[43]. This requires phase separation within the amorphous phase, which can be seen in Fig. 3d where Cu precipitates were observed in direct contact with $\alpha$-Fe(Si) grains in a sample annealed at 723 K. The observed increase in cluster size in conjunction with a decrease in number density suggests the occurrence of the classic Ostwald ripening phenomenon (Supplementary Fig. 4). Thus, the microstructure transformation in the Nanomet melt-spun ribbon occurs through a series of three distinct stages: (1) an initial amorphous phase, (2) an amorphous phase with the formation of clusters, and (3) nucleation of an $\alpha$-Fe(Si) phase at the heterogeneous sites of clusters trailed by subsequent grain growth, reduction in the amorphous phase and the appearance of additional Fe-B compound phases.

## Thermal transport properties of nanostructure-engineered Fe-based ribbons

The thermal transport properties of the nanostructure-engineered Nanomet ribbons were investigated. Figure 4a displays the thermal conductivity $\kappa$ of the annealed samples, estimated via $\kappa = \rho_d C_p D$ using the thermal diffusivity $D$, specific heat $C_p$, and material density $\rho_d$ (Supplementary Figs. 5–8). $D$ was measured using a laser-spot-periodic-heating method based on lock-in thermography (LIT) (see "Methods" section and Supplementary Figs. 5–7)[44]. No significant change in the $D$ values was observed for the samples annealed below the onset of crystallization temperature ($T_{x1}$ = 661 K) and the values were found to be in a range of 2.2–2.6 × $10^{-6}$ m$^2$s$^{-1}$. A steep rise in $D$ was observed as the crystallization of $\alpha$-Fe(Si) started in the amorphous matrix for samples annealed at 673 K and 723 K, with an average value of 3.5 × $10^{-6}$ m$^2$s$^{-1}$ and 4.1 × $10^{-6}$ m$^2$s$^{-1}$, respectively (Supplementary Fig. 6). $C_p$ did not reveal any discernible patterns, with values ranging between 0.47 and 0.56 Jg$^{-1}$K$^{-1}$ (Supplementary Fig. 8), closely resembling the specific heat of pure Fe. Figure 4a shows that $\kappa$ increases with the annealing temperature, with a sudden rise in its value after 673 K. Thus, the observed variations in $\kappa$ are primarily attributed to the changes in $D$, while $C_p$ remains nearly constant throughout the annealing process. The longitudinal electrical conductivity $\sigma$ of the annealed samples was also studied to estimate the contribution of phonons and electrons in $\kappa$ through the Wiedemann-Franz law[45]. Figure 4b indicates that $\sigma$ exhibits a similar trend to $\kappa$ with a substantial

increase beyond the crystallization temperature. The electron and phonon thermal conductivities were respectively estimated as $\kappa_e = \sigma L T$ and $\kappa_p = \kappa - \kappa_e$, with $L$ being the Lorenz number (2.44 × $10^{-8}$ W$\Omega$K$^{-2}$) and $T$ the absolute temperature (300 K; Supplementary Fig. 9). Interestingly, phonons and electron have a comparable contribution to the thermal conductivity for the samples annealed below 643 K. However, a slight dominance of the phonon contribution was observed for the sample containing a high-density of Cu-clusters (annealed at 653 K) and for the samples annealed beyond the crystallization temperature. Hence, $\kappa$ and $\sigma$ behaviors can be attributed to the well-ordered atomic structure in the crystalline materials, which facilitates efficient phonon/electron transport and thus leads to higher conductivities. In contrast, the amorphous phase, characterized by structural disorder and more scattering sites, hinders phonon/electron transport, and resulting in the smaller $\kappa_e$ and $\kappa_p$ values. It is worth noting that in our samples, the formation of clusters increases $\kappa$ due to the presence of Cu-rich clusters in the amorphous matrix, while the presence of clusters can also affect these properties by introducing additional scattering sites for both phonons and electrons. Therefore, the formation of a high-density Cu-clusters in the amorphous matrix facilitates the electron and phonon transport by reducing the volume fraction of the disordered structure. In the future, the phonon engineering approach can be implemented to fine-tune the transport properties[46].

## Large anomalous Nernst effects in nanostructure-engineered Fe-based amorphous ribbons

To investigate the impact of microstructure tailoring on the transverse thermoelectric properties, we studied AEE using the thermoelectric imaging technique based on LIT method. This technique allowed us to examine the spatial distribution of temperature modulation and symmetry of AEE. Steady-state AEE signals were analyzed to quantitatively estimate the anomalous Ettingshausen coefficient ($\Pi_{AEE}$) at room temperature ($T$ = 300 K) (see Fig. 4c, "Methods" section, and Supplementary Fig. 10). The corresponding $S_{ANE}$ values of the samples were determined through the Onsager reciprocal relation: $S_{ANE} = \Pi_{AEE}/T \equiv \Pi_{AEE}/300$, also presented in Fig. 4c. The magnitude of $S_{ANE}$ for the as-quenched sample was estimated to be 2.2 µVK$^{-1}$. This value is already higher than that for the other polycrystalline Fe-based alloys and an order of magnitude larger than that for pure Fe[2,12]. During annealing, the $S_{ANE}$ value remained relatively stable up to 623 K. It increased to 2.7 µVK$^{-1}$ for the sample annealed at 643 K and surprisingly shoots up to a

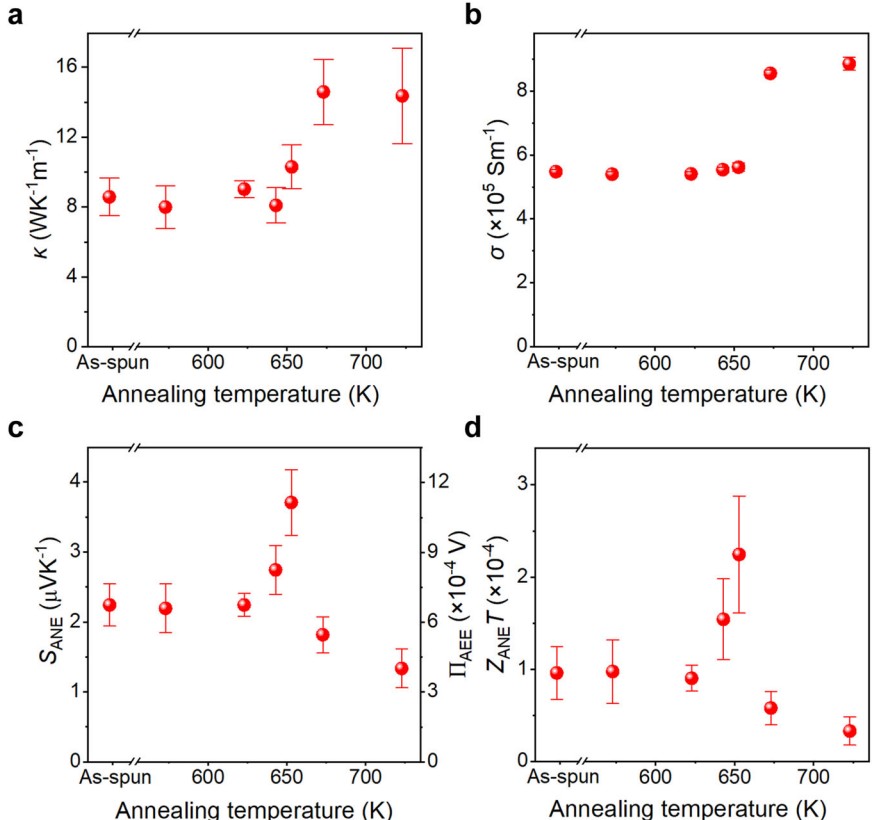

**Fig. 4 | Transport properties of annealed Nanomet samples. a–d** Annealing temperature dependence of the thermal conductivity $\kappa$ (**a**), electrical conductivity $\sigma$ (**b**), anomalous Nernst coefficient $S_{ANE}$ and the corresponding anomalous Ettingshausen coefficient $\Pi_{AEE}$ estimated using the Onsager reciprocal relation at 300 K (**c**), and dimensionless figure of merit for ANE $Z_{ANE} T$ at 300 K (**d**) for the Nanomet samples. The error bars represent the standard deviation of the measurements.

remarkably higher value of 3.7 μVK⁻¹ for the sample annealed at 653 K, which is 70% higher than that for the as-quenched sample. However, when the annealing temperature exceeds 673 K and the crystallization of α-Fe(Si) proceeds, the $S_{ANE}$ values decrease. The substantial increase in $S_{ANE}$ was only observed for the sample with a high-density of Cu-enriched clusters within the amorphous matrix. This could be attributed to the interfacial spin-orbit interaction at the boundaries between the Fe-based amorphous matrix and the Cu-clusters[47]. In addition, the crystallization of α-Fe(Si) leads to a compositional change in the amorphous phase, which could be optimized in the fully amorphous samples with large ANE. Although the formation of Cu-clusters in the amorphous matrix resulted in the high $S_{ANE}$ value, theoretical studies are desirable to elucidate the microscopic origin[28].

## Discussion

$S_{ANE}$ can be described as two components ($S_I$ and $S_{II}$) using the linear response equation[17]: $S_{ANE} = \rho_{xx}\alpha_{xy} - \rho_{AHE}\alpha_{xx} \equiv S_I + S_{II}$. Here, $\rho_{xx}$, $\rho_{AHE}$, $\alpha_{xy}$, and $\alpha_{xx}$ represent the longitudinal electrical resistivity, anomalous Hall resistivity, transverse thermoelectric conductivity, and longitudinal thermoelectric conductivity, respectively. The intrinsic ANE that directly converts temperature gradient into a transverse voltage is signified by the component $S_I$, whereas the $S_{II}$ reflects the voltage originating from the Hall effect of the longitudinal charge flow induced by the Seebeck effect (SE) due to the anomalous Hall effect (AHE). The $S_{II}$ can be rewritten as $\frac{-S_{xx}\rho_{AHE}}{\rho_{xx}} = -S_{xx}\tan\theta_{AHE}$, where $S_{xx}$ is the Seebeck coefficient and $\theta_{AHE}$ is the anomalous Hall angle. To discern the $S_I$ and $S_{II}$ contribution to ANE, we estimated the $\tan\theta_{AHE}$ and $\alpha_{xy}$ by conducting transverse electric resistivity ($\rho_{yx}$) and $S_{xx}$ measurements at room temperature for the Nanomet samples annealed at 643 K and 653 K. This enables us to gain better understanding on the origin of an

increase in $S_{ANE}$. Table 1 illustrated the estimated values, revealing that the magnitude of $S_{II}$ in both samples is considerably smaller compared to $S_I$. This infers that ANE is primarily governed by the $S_I$ components due to the large value of $\alpha_{xy}$, while the impact of the anomalous Hall and Seebeck effects is negligible. The $\alpha_{xy}$ value for the Nanomet samples annealed at 643 K and 653 K was 1.41 and 1.93 Am⁻¹K⁻¹, respectively. This significant difference in $\alpha_{xy}$ values is consistent with the trend observed in $S_{ANE}$ and can be attributed to the presence of a high density of nano-size Cu clusters in amorphous matrix.

In Fig. 5a, b, we respectively compared the $|S_{ANE}|$ and power factor (PF) $\sigma S_{ANE}^2$ values for various spin-caloritronic amorphous materials reported in the literature with the Fe-based amorphous materials developed in this study. Notably, the Nanomet sample annealed at 653 K exhibits a significantly large $S_{ANE}$ value (~3.7 μVK⁻¹) and a high PF (~7.7 μWm⁻¹K⁻²) at room temperature, which is larger than the values observed in all of the conventional spin-caloritronic amorphous materials and comparable to the single-crystalline materials reported so far[2,8,10,30]. Furthermore, it is important to highlight that most of the bulk polycrystalline materials exhibit $|S_{ANE}|$ values of <1 μVK⁻¹, while only a few materials possess $S_{ANE}$ of >3 μVK⁻¹ at room temperature[12,30], such as the SmCo₅ (~3.5 μVK⁻¹)[32,33] and Co₂MnGa (6–8 μVK⁻¹) systems[7,8,17,19] (Supplementary Fig. 11). However, both are expensive due to the existence of Co or Sm and are not flexible. Therefore, it is worth noting that our low-cost and flexible material demonstrates the high $S_{ANE}$ value at room temperature, achieved through nanostructure engineering. Figure 5c highlights the mechanical flexibility of the Nanomet materials below the onset crystallization temperature. This figure vividly demonstrates their remarkable ability to be easily curved into various shapes, confirming their suitability for practical applications that demand flexibility. This nanostructure engineering approach

**Table 1 | The anomalous Hall resistivity $\rho_{AHE}$, $\tan\theta_{AHE}$, transverse thermoelectric conductivity $\alpha_{xy}$, Seebeck coefficient $S_{xx}$, $S_I$, and $S_{II}$ values at room temperature for the Nanomet samples annealed at 643 K and 653 K**

| Sample | $\rho_{AHE}$ ($\times 10^{-8}$ $\Omega m$) | $\tan\theta_{AHE}$ ($\times 10^{-2}$) | $\alpha_{xy}$ ($Am^{-1}K^{-1}$) | $S_{xx}$ ($\mu VK^{-1}$) | $S_I$ ($\mu VK^{-1}$) | $S_{II}$ ($\mu VK^{-1}$) |
|---|---|---|---|---|---|---|
| Annealed at 643 K | 7.80 | 4.31 | 1.41 | −4.52 | 2.54 | 0.20 |
| Annealed at 653 K | 7.16 | 4.02 | 1.93 | −6.48 | 3.44 | 0.26 |

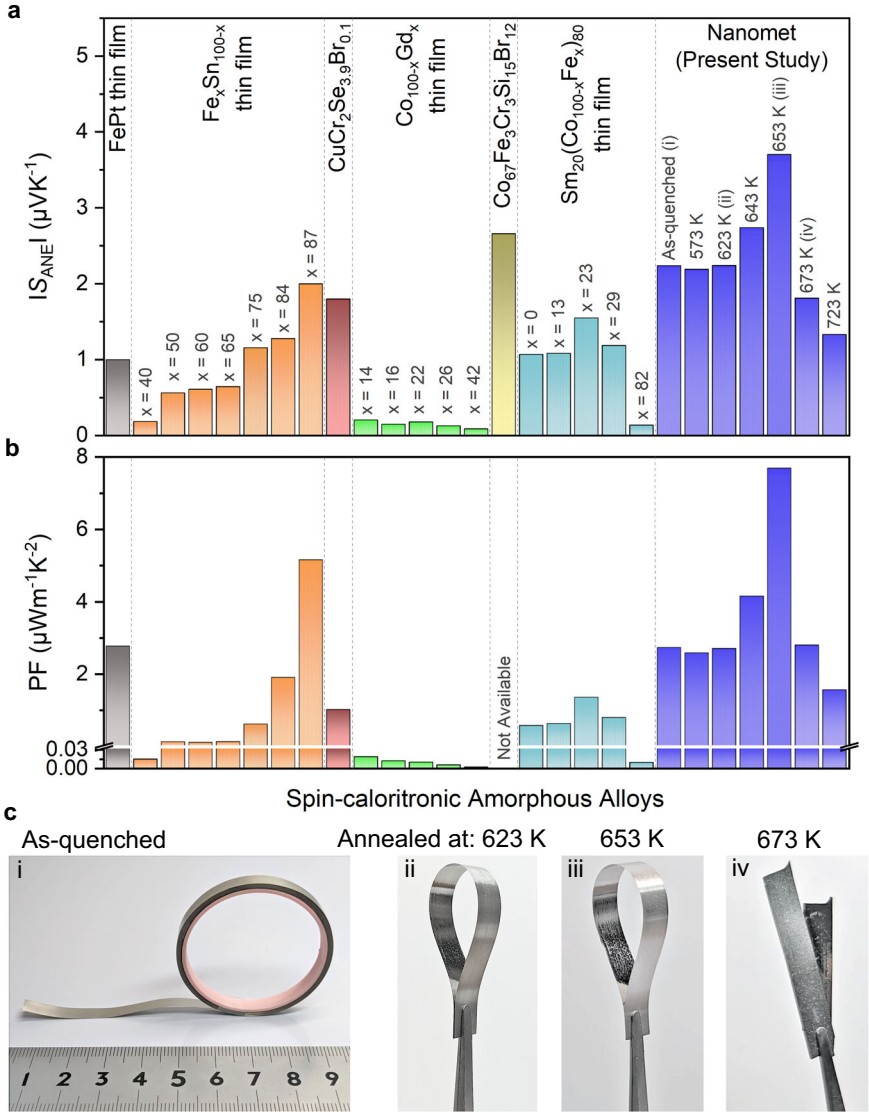

**Fig. 5 | Spin-caloritronic amorphous alloys for the anomalous Nernst effect and demonstration of mechanical flexibility of Nanomet samples.** Comparison of **a**, the absolute values of $S_{ANE}$ and **b**, power factor for various spin-caloritronic amorphous alloys measured at room temperature. Notably, the Nanomet ribbon (present study) annealed at 653 K exhibits the significantly large $|S_{ANE}|$ and power factor value compared to that for the other reported alloys (Fe-Pt[37], $Fe_xSn_{100-x}$[28], $CuCr_2Se_{3.9}Br_{0.1}$[40], $Co_{100-x}Gd_x$[36,39], $Co_{67}Fe_3Cr_3Si_{15}Br_{12}$[38], and $Sm_{20}(Co_{100-x}Fe_x)_{80}$[34]). **c** Highlighting the mechanical flexibility of long, thin Nanomet ribbons prepared in different conditions: **c.i.** as-quenched, **c.ii.** annealed at 623 K, and **c.iii.** annealed at 653 K. However, **c.iv.** reveals the brittleness of the ribbon annealed at 673 K, as it broke when subjected to bending.

enables the enhancement of $S_{ANE}$ within the same materials, leading to elevated levels of performance.

To assess the effectiveness of the Nanomet ribbons as spin-caloritronic materials for ANE/AEE applications, we estimated the dimensionless figure of merit for ANE through the equation[33]: $Z_{ANE}T = \frac{S_{ANE}^2 \sigma}{\kappa} T$. As shown in Fig. 4d, the maximum $Z_{ANE}T$ value was found to be $2.2 \times 10^{-4}$ at $T = 300$ K for the sample annealed at 653 K, which is three orders of magnitude larger than that for pure Fe and comparable to the record-high $S_{ANE}$ values at room temperature for the $SmCo_5$-type permanent magnets ($4.5 \times 10^{-4}$)[32,33] and $Co_2MnGa$

Heusler ferromagnet ($2.0 \times 10^{-4}$)[7,8,48]. This indicates that the Nanomet ribbons with embedded nano-sized Cu-clusters in an amorphous matrix can offer an excellent combination of microstructure features for enhancing the transverse thermoelectric conversion properties. From Fig. 4, it can be inferred that the large value of $S_{ANE}$ is the dominant factor contributing to the improvement of $Z_{ANE}T$. However, it is still insufficient for practical applications. Therefore, $S_{ANE}$ and $Z_{ANE}T$ are necessary to be further improved by tailoring the size, distribution, and composition of the clusters and composition of the matrix to achieve the desired functionalities. Larger $Z_{ANE}T$ can also be

obtained by reducing $\kappa_p$ through phonon engineering. Furthermore, we validated our nanostructure engineering approach in tailoring ANE using another Fe-based amorphous material with a higher Cu content (Supplementary Figs. 12–15). Increase of Cu content in the alloy composition to 1.5 at% led to an increase in the PF to ~8.9 $\mu Wm^{-1}K^{-2}$ and $Z_{ANE}T$ to ~2.5 × $10^{-4}$ (see Supplementary Fig. 15).

In conclusion, we propose and demonstrate a novel approach to design low-cost and flexible spin-caloritronic materials that exhibit large ANE through nanostructure engineering. This approach is of great importance for advancing the use of ANE in thermoelectric devices. These materials have immense potential in the fabrication of highly efficient and flexible energy harvesting and thermal management devices with curvilinear design. This study has established a direct correlation between ANE and engineered nanostructures in a spin-caloritronic material. As a result, we have discovered a new methodology for tailoring ANE using nanoscale clusters embedded in an amorphous matrix microstructure, which is a proof-of-concept for future advancements. The proposed method is applicable to various systems via the development of nanocomposite materials, which can open up a new avenue towards the development of spin-caloritronic materials with giant ANE suitable for practical applications.

## Methods

### Preparation of Fe-based amorphous alloy
The master alloy ingot was prepared by melting high-purity elements of Fe (99.99%), Si (99.99%), Cu (99.9%), B (99.5%), and $Fe_3P$ (99%) using the vacuum induction melting technique under an Ar atmosphere. Subsequently, the amorphous ribbon was produced using the single-roll melt-spinner. The ingot was melted in a quartz tube and then ejected under an Ar atmosphere at a pressure of 0.02 MPa onto a rotating copper (Cu) wheel. The tangential speed of the wheel was set at 35 m $s^{-1}$, while the gap between the quartz tube and the rotating Cu wheel was maintained at 0.2 mm. The process flow chart was optimized to obtain high-quality amorphous ribbons with lengths of several meters, a width of 5 mm, and a thickness ranging from 16 to 20 μm, as shown in Fig. 5c.

For annealing, the samples were placed in quartz tubes and connected to a high vacuum turbo pump to avoid oxidation. To achieve rapid annealing and to prevent undesired microstructural changes during heating, the samples were directly inserted into a preheated tubular furnace. Annealing was performed at different temperatures of 573, 623, 643, 653, 673, and 723 K for a soaking time of 3 min. The temperature of the ribbons was continuously monitored during annealing using a K-type thermometer positioned in close proximity to the ribbons. Notably, a heating rate of nearly 120 K $s^{-1}$ was attained, enabling precise temperature control. Further, the ribbons were allowed to cool naturally to room temperature by removing the quartz tube from the furnace.

### Characterization of samples
The chemical composition was estimated using inductively coupled plasma optical emission spectrometry (ICP-OES) and was found to be $Fe_{84.7}Si_{2.8}P_{3.8}B_{7.8}Cu_{0.7}C_{0.2}$ (at%). This composition is associated with the trademark "Nanomet". The ribbon was cut into rectangular samples with the dimensions of ~ 60 × 5, ~ 10 × 5, and ~ 10 × 1 ± 0.1 mm for $\sigma$ and $D$, and AEE measurements, respectively. The thermal analysis of the as-quenched ribbon was carried out by differential scanning calorimetry (DSC, Rigaku TG8120) at a heating rate of 20 K/min in an Ar atmosphere. The phase analysis of the annealed samples was evaluated by XRD (Rigaku MiniFlex600) with Cr-Kα radiation. Microstructural studies were carried out using a transmission electron microscope (TEM, FEI Titan $G^2$ 80-200 equipped with a probe corrector). The elemental distribution was investigated using atom probe tomography (APT) in the laser mode, utilizing the CAMECA LEAP 5000 XS instrument operating at a repetition rate of 250 kHz. The laser pulse energy

was set at 30 pJ, and the experiments were carried out at a base temperature of 30 K, maintaining a constant detection rate of 1.5%. The obtained APT data was subsequently analyzed using CAMECA AP Suit 6.1 software. The TEM specimens and APT tip were prepared by lift-out and annular milling techniques using a dual beam focused ion beam (FEI Helios 5UX).

### Transport properties measurement
The angular distribution of the in-plane $D$ was measured by means of the LIT method. A schematic of the setup is plotted in Supplementary Fig. 5 comprises an infrared camera, a diode laser, a function generator, and a LIT system. The diode laser emits a modulated laser beam driven by a periodic reference signal at frequency $f$ from the function generator. The modulated beam was focused as a point on the backside of the opaque sample by an optical setup. The laser heat-point in turn induces in-plane heat waves that diffuse radially within the sample. Simultaneously, the camera images the thermal response on the front side of the sample. The LIT system processes the thermal images according to $f$ and outputs the spatial distribution of the lock-in amplitude $A$ and phase $\phi$. $D$ can be estimated from the correlation between $\phi$ and the distance $r$ to the periodic point-heat source within the circular diffusion area. Hence, the angular distribution of $D$ can be obtained by revolving the analysis around the heating point at angles $\theta$ (Supplementary Fig. 5)[49].

$$D = \frac{\pi f}{(d\phi/dr)^2} \qquad (1)$$

The measurement was performed with a science-grade LIT system (ELITE, DCG Systems Inc.) and a diode laser of 638 ±1 nm wavelength (LDM637D.300.500, Omicron Inc.). The laser heat spot is focused to a diameter of ~7 μm at a power $P$ = 10 mW and modulated at $f$ = 3 Hz. $f$ and $P$ are selected by taking into consideration maximizing the signal-to-noise ratio, reducing the heat loss effect[50], and the thermal diffusion length $\Lambda = \sqrt{D/\pi f}$ does not exceed the sample width[49]. The $C_p$ was measured by differential scanning calorimetry (DSC, Rigaku Thermo plus EV02). The $\kappa$ at room temperature was estimated as

$$\kappa = C_p \rho D, \qquad (2)$$

where $\rho$ is the density of the Nanomet ribbons which is around 7.46 g $cm^{-3}$.

AEE of all the samples was examined also by means of the LIT method[12]. The rectangular Nanomet samples with the dimensions of ~ 10 × 1 ± 0.1 $mm^2$ were used to measure AEE. The sample was fixed on a glass substrate to reduce the heat loss from the sample to the sample stage because of the low thermal conductivity of glass. The two wires (left- and right-side in Supplementary Fig. 10) were connected in series and directed the charge current in opposite directions between the wires. The thermal images of the surface of the samples were obtained by applying a square-wave-modulated periodic charge current with amplitude $J_c$, frequency $f$, and zero offset to the ribbons in the x-direction with an applied magnetic field, $\mu_0 H$ along the y-direction (Supplementary Fig. 10). The first harmonic response of the detected images was extracted and transformed into $A$ and $\phi$ images through Fourier analysis. Using this methodology, it is possible to isolate and identify the sole effect of thermoelectric effects ($\propto J_c$) without any interference from Joule heating ($\propto J_c^2$)[32,51,52]. In the LIT-based thermoelectric measurements, the $A$ image represents the magnitude of current-induced temperature modulation and the $\phi$ image the sign, that is, $\phi$ ~ 0° (-180°) corresponds to releasing (absorbing) heat, as well as the time delay of the temperature modulation. The LIT measurements were performed by applying $J_c$ = 1.0 A at room temperature ($T$ = 300 K), $\mu_0 H$ = ± 1 T, and $f$ = 1.0-10.0 Hz. Since the AEE-induced temperature modulation shows the H-odd dependence, we calculated

the $H$-odd component of lock-in amplitude $A_{\text{odd}}$ and phase $\phi_{\text{odd}}$ by $A_{\text{odd}} = |A(+H)e^{-i\phi(+H)} - A(-H)e^{-i\phi(-H)}|/2$ and $\phi_{\text{odd}} = -\arg[A(+H)e^{-i\phi(+H)} - A(-H)e^{-i\phi(-H)}]$, where $A(+H)$ $[\phi(+H)]$ and $A(-H)$ $[\phi(-H)]$ show the $A$ $(\phi)$ value measured at $\mu_0 H = +1\,\text{T}$ and $-1\,\text{T}$, respectively. The AEE-induced temperature modulation at the steady state $A_{\text{odd}}^{\text{steady}}$ was determined by using the magnitude of $A_{\text{odd}}$ signals at $f = 1.0\,\text{Hz}$ because of their $f$-independence in the low $f$ region. The steady-state AEE signals were analyzed to quantitatively estimate the $\Pi_{\text{AEE}}$ values at 300 K using the equation[33]:

$$\Pi_{\text{AEE}} = \pi \kappa |\Delta T_{\text{AEE}}| / 4 j_c t, \qquad (3)$$

where $\Delta T_{\text{AEE}}$ represents the temperature change induced by AEE between the top and bottom surfaces of the samples, determined using $|\Delta T_{\text{AEE}}| = 2 A_{\text{odd}}^{\text{steady}}$, $j_c$ the charge current density, and $t$ the thickness of the samples. To estimate the figure of merit, $\sigma$ was measured using a four-probe method.

For measurements of $\rho_{\text{AHE}}$ and $S_{xx}$, ribbon samples, with a width of 5 mm, a length of 10 mm, and a thickness of 12 μm, were fixed on a glass substrate. To estimate $\rho_{\text{AHE}}$, the value of transverse electric resistivity $\rho_{yx}$ was determined through the Hall measurement using a physical property measurement system (PPMS, Quantum Design, Inc.), where a charge current of 0.1 A was applied along the longitudinal direction (Supplementary Fig. 16). The $\rho_{\text{AHE}}$ value was determined by extrapolating the slope of the high-field data to zero-field. The $S_{xx}$ value was measured using the Seebeck Coefficient/Electric Resistance Measurement System (ZEM-3, ADVANCE RIKO, Inc.).

## Reporting Summary

Further information on research design is available in the Nature Portfolio Reporting Summary linked to this article.

## Data availability

The data that support the findings of this study are available from the corresponding authors on reasonable request.

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

## Acknowledgements

The authors thank Y. Sakuraba and K. Hono for valuable discussions and W. Zhou, M. Isomura, H. Sebata, and N. Kurata for technical support. This work was supported by JST ERATO "Magnetic Thermal Management Materials" (Grant No. JPMJER2201).

## Author contributions

R.G. and H.S.A. conceived the idea, planned, and conducted the microstructural characterization. R.G. designed the experiments, prepared the samples, collected, and analysed the data. R.G., H.S.A. and T.O. conducted atom probe tomography experiments. K.U. conducted AEE measurements and T.H. analysed the AEE data. T.H. performed measurements of AHE and Seebeck effect. A.A. and H.N. measured the thermal diffusivity. H.S.A. and K.U. supervised the study. All authors discussed the results and contributed to the preparation and revision of the manuscript.

## Competing interests

The authors declare no competing interests.
