## [Peer Review File · Nature Communications]

Creation of flexible spin-caloritronic material with giant transverse thermoelectric conversion by nanostructure engineeringReviewers' comments:

Reviewer #1 (Remarks to the Author):

This work reported a significant improvement in the anomalous Nernst coefficients (ANE) in Fe-based amorphous materials through nanostructure engineering. In addition, ANE may be used for transverse thermoelectric devices, so this is an interesting research topic. The experiments were well designed and conducted, and the manuscript has also been well prepared. I like to recommend it for publication in Nature Communications after minor revisions considering my following comments:

- (1) I feel the abstract is not specific, in particular the key results obtained in this study as summarized in figures and text.
- (2) The authors use the term of “spin-caloritronic” in the title and emphasize “This discovery advances the materials science of spin-caloritronics” in the abstract. I am not familiar with it, so I think the authors may need write some words about it to make connection with the present investigated materials in the text or in the introduction.
- (3) Is there any other materials that show larger SANE and PF except those shown in Figure 5?
- (4) Please consider to unify the usage of “Structure engineering and structural engineering” in the whole manuscript.

Reviewer #2 (Remarks to the Author):

The manuscript entitled Creation of flexible spin-caloritronic material with giant transverse thermoelectric conversion by nanostructure engineering submitted by R. Gautam et al. reported the synthesis of Fe-based magnetic materials by nanostructure engineering. In this work the large anomalous Nernst effect was observed. The high performance of ANE results from the optimization of microstructures for the samples. Furthermore, thermal and electrical transport properties were discussed in detail. Unfortunately, the so-called “nanostructure engineering” method to fabricate samples is not a new one but well-known crystallization of amorphous solid technique, which has been developed more than 30 years ago. In addition, the physical mechanism of ANE improvement with specific annealing temperature and structure is not fully clarified. Based on above reasons, I am inclined to conclude that the current manuscript is NOT suitable for publication in Nature Communications. Besides, some comments questions are listed as following:

- 1) It is written that “...resulted in the figure of merit for ANE comparable to the record-high value in a Heusler ferromagnet at room temperature...” “... larger than the values observed in all of the conventional spin-caloritronic amorphous materials and comparable to the single-crystalline materials reported so far...” “...only a few materials, such as the SmCo₅ and Co₂MnGa systems, possess SANE of >3 μ VK⁻¹ at room temperature”
Therefore, it is necessary to list the specific value or graphically display the related value and make comparisons.
- 2) It is found that the formation of Cu-cluster enhances ANE with annealing temperature lower than 653 K. Why ANE decreases with cluster growing further?
- 3) The ANE enhancement was attributed the interfacial spin-orbit interaction at the boundaries between the Fe-based amorphous matrix and the Cu-clusters. Considering that there is no heavy elements such as Pt (PRB 91(2015)045407) but 3d elements Cu and Fe, how spin-orbit interaction occurs?
- 4) Line 218. SAAE should be written as SANE.

- 5) Line 211. Since T refers to 300 K, T is suggested to be replaced by 300 K directly. Otherwise, T may be confused with annealing temperatures especially in Fig. 4c.
- 6) It can be seen from Fig. 4c and Fig. 4d that the error bars are obvious. So the tendency of variation for different samples is not convincing. It is better to add one sample with the annealing temperature around 663 K.

Reviewer #3 (Remarks to the Author):

The paper by R. Gautam et al. presents an interesting improvement of the anomalous Nernst effect (ANE) in flexible Fe-based amorphous materials through nonstructural engineering. However, because of the presence of some deficient in the results, we suggested that the manuscript in the present form is not suitable for publication in the journal of Nature Communications. The reason is as follows:

- i) Though the authors give a analysis of the evolution of the Cu nano-clustering with annealing temperature by TEM and APT, to understand the improvement of the ANE in detail, the particle size of the Cu clustering and its content in the composite should be quantitatively estimated.
- ii) To estimate the contribution of Cu nano-clustering in the thermal electric effect in the Fe based nanomet, the thermal conductivity and ANE effect of the pure Cu particle are required.
- iii) To improve the level of the manuscript, a theoretical analysis of the improvement of the ANE effect with the nonstructural engineering is highly desired.

Based on the above points, the manuscript may not meet the level for publication in this journal.

Response Letter

We appreciate the reviewers for reading our manuscript and giving us constructive comments. We incorporated all the comments in the revised version, which are indicated in red in the submitted manuscript. How we revised the manuscript based on the comments are listed below:

Response to Reviewer 1

<Reviewer 1>: This work reported a significant improvement in the anomalous Nernst coefficients (ANE) in Fe-based amorphous materials through nanostructure engineering. In addition, ANE may be used for transverse thermoelectric devices, so this is an interesting research topic. The experiments were well-designed and conducted, and the manuscript has also been well-prepared. I like to recommend it for publication in Nature Communications after minor revisions considering my following comments: I feel the abstract is not specific, in particular the key results obtained in this study as summarized in figures and text.

<Reply>: Following the reviewer's suggestions, the abstract is revised by incorporating the key results obtained in this study. These points are addressed in the abstract as follows:

“We demonstrate a remarkable ~70% improvement in the anomalous Nernst coefficients (reaching ~3.7 μVK^{-1}) and a significant ~200% enhancement in the power factor (reaching ~7.7 $\mu\text{Wm}^{-1}\text{K}^{-2}$) in flexible Fe-based amorphous materials by nanostructure engineering without changing their composition.”

<Reviewer 1>: The authors use the term of “spin-caloritronic” in the title and emphasize “This discovery advances the materials science of spin-caloritronics” in the abstract. I am not familiar with it, so I think the authors may need write some words about it to make connection with the present investigated materials in the text or in the introduction.

<Reply>: We apologize for the lack of explanation. We have explained the term of "spin-caloritronics" in the introduction section of the revised manuscript as follows:

“These devices leverage thermoelectric effects, enabling the direct interconversion between heat and electricity. Additionally, heat currents can also interact with spin-dependent transport, giving rise to the field of spin-caloritronics, which focuses on the interconversion of spin, charge, and heat currents³⁻⁵.”

<Reviewer 1>: Is there any other materials that show larger SANE and PF except those shown in Figure 5?

<Reply>: Figure 5 includes the S_{ANE} and PF values for all the reported spin-caloritronic amorphous alloys. In addition to these materials, we have highlighted the values of S_{ANE} and figure of merits for the polycrystalline materials (SmCo₅ and Co₂MnGa systems) on pages 11 and 12 which exhibit record-high S_{ANE} values.

<Reviewer 1>: Please consider to unify the usage of “Structure engineering and structural engineering” in the whole manuscript.

<Reply>: Thank you for your comment on the consistency in the use of terminology throughout the manuscript. We have carefully reviewed the text to ensure consistency in referring to "Structure engineering" and highlighted the changes in the revised manuscript.

Response to Reviewer 2

<Reviewer 2>: The manuscript entitled Creation of flexible spin-caloritronic material with giant transverse thermoelectric conversion by nanostructure engineering submitted by R. Gautam et al. reported the synthesis of Fe-based magnetic materials by nanostructure engineering. In this work the large anomalous Nernst effect was observed. The high performance of ANE results from the optimization of microstructures for the samples. Furthermore, thermal and electrical transport properties were discussed in detail. Unfortunately, the so-called “nanostructure engineering” method to fabricate samples is not a new one but well-known crystallization of amorphous solid technique, which has been developed more than 30 years ago. In addition, the physical mechanism of ANE improvement with specific annealing temperature and structure is not fully clarified. Based on above reasons, I am inclined to conclude that the current manuscript is NOT suitable for publication in Nature Communications. Besides, some comments questions are listed as following:

<Reply>: We would like to thank the reviewer for providing valuable input and critical feedback on the manuscript. We fully agree with the reviewer that the concept of crystallization from an amorphous matrix is a well-known phenomenon. However, one of the main achievements and claim of this paper is that unlike the conventional approach which has been focused on increasing ANE by materials design and Berry curvature engineering, nanostructure engineering can lead to an increase in ANE, which has never been reported. Therefore, our manuscript introduces a new dimension by focusing on microstructure engineering to advance the ANE. Specifically, we have tailored the microstructure at different length scales and highlighted its significant impact on transport properties and ANE.

Our focus on targeted microstructural engineering involves the formation of nanoscale Cu clusters (~2 nm in size) embedded in the amorphous matrix. This approach resulted in a remarkable 70% improvement in ANE compared to the material with a fully amorphous or polycrystalline structure. Notably, we have also developed a spin-caloritronic material that is highly flexible, which is very important for practical applications. We strongly believe that our microstructure engineering strategy, as explained in the manuscript, represents a significant advancement in the field and paves the way for a new approach to enable efficient transverse thermoelectric conversions.

In addition, we greatly appreciate the reviewers' comments on the physical mechanism of the S_{ANE} improvements according to the nanostructure changes. We have addressed this point by adding a new section in the manuscript on page 10-11 and measuring the intrinsic and extrinsic contributions to S_{ANE} . Based on these new clarifications and the incorporation of all the points raised by the reviewer, we trust that the revised manuscript is worthy of further consideration.

<Reviewer 2>: It is written that “...resulted in the figure of merit for ANE comparable to the record-high value in a Heusler ferromagnet at room temperature...” “... larger than the values observed in all of the conventional spin-caloritronic amorphous materials and comparable to the single-crystalline materials reported so far...” “...only a few materials, such as the SmCo5 and Co2MnGa systems, possess S_{ANE} of $>3 \mu\text{VK}^{-1}$ at room temperature”. Therefore, it is necessary to list the specific value or graphically display the related value and make comparisons.

<Reply>: We appreciate the reviewer raising this point. We have highlighted the values of S_{ANE} and figure of merits for the polycrystalline materials (SmCo5 and Co2MnGa systems) on page 11 and 12 which exhibit record-high S_{ANE} values. In addition to this, we have also included a comparison graph in supplementary Figure S11 to address the reviewer's comment. In fact, the reviewer points out another important achievement of this work. Although only a few materials such as SmCo5 and Co2MnGa systems have S_{ANE} of $>3 \mu\text{VK}^{-1}$ at room temperature, we show for the first time that a large S_{ANE} can also be realized by nanostructure engineering.

Fig. S11 | Comparison of the anomalous Nernst effect for spin-caloritronic materials. Comparison of the absolute values of the anomalous Nernst coefficient ($|S_{ANE}|$) with the magnetization ($\mu_0 M_s$) for various bulk spin-caloritronic materials measured at room temperature¹⁰. The green dotted line represents the upper limit of the $|S_{ANE}|$ signals in conventional materials.

<Reviewer 2>: It is found that the formation of Cu-cluster enhances ANE with an annealing temperature lower than 653 K. Why ANE decreases with cluster growing further?

<Reply>: The enhancement of ANE was indeed associated with the formation of nanoscale Cu clusters during annealing at temperatures below 653 K. As the temperature is increased beyond 653 K, approaching the onset of crystallization temperature (661 K according to DSC), we observed the crystallization of the α -Fe phase at the heterogeneous sites of the Cu clusters. Thus, the growth of the cluster size is accompanied with crystallization of α -Fe. Fe is known to have a negative anomalous Nernst coefficient, which offsets the positive anomalous Nernst coefficient in other phases. Therefore, all these factors including growth of Cu cluster size, reduction of their density, and nucleation of α -Fe crystals are responsible for decrease of ANE. These microstructural factors can potentially alter the electron scattering mechanisms or inducing changes in the electronic band structure resulting in a decrease in S_{ANE} .

Fig. S4 | Variation in the number density and size of Cu clusters for Nanomet samples as a function of annealing temperature. The white region indicates the presence of Cu-clusters in an amorphous matrix, while the grey dotted line shows the onset of primary crystallization and grey regions signify the presence of α -Fe crystalline phase and Cu-clusters in an amorphous matrix. The red dotted line indicates the annealing temperature (653 K) that exhibits the highest S_{ANE} value.

<Reviewer 2>: The ANE enhancement was attributed the interfacial spin-orbit interaction at the boundaries between the Fe-based amorphous matrix and the Cu-clusters. Considering that there is no heavy elements such as Pt (PRB 91(2015)045407) but 3d elements Cu and Fe, how spin-orbit interaction occurs?

<Reply>: Heavy elements are not always necessary to increase ANE. As demonstrated, for example, in "Phys. Rev. B **92**, 094414 (2015)," increased ANE in magnetic multilayers has been observed not only in systems containing noble metals, such as Fe/Pt and Fe/Au multilayers, but also in Fe/Cu multilayers. Although the microscopic origin of this enhancement is still under debate, alloying in a few atomic layers at the interfaces and enhancement of the interfacial spin-orbit coupling have been discussed. Also, as reported in "Appl. Phys. Express **13**, 043001 (2020)" and "Nature **581**, 53 (2020)", large ANE has been observed in binary alloy systems that do not contain heavy elements.

<Reviewer 2>: Line 218. S_{AEE} should be written as S_{ANE} .

<Reply>: We apologize for the mistake. It is corrected in the revised manuscript.

<Reviewer 2>: Line 211. Since T refers to 300 K, T is suggested to be replaced by 300 K directly. Otherwise, T may be confused with annealing temperatures especially in Fig. 4c.

<Reply>: As suggested by the reviewer, we have corrected this in the revised manuscript.

<Reviewer 2>: It can be seen from Fig. 4c and Fig. 4d that the error bars are obvious. So the tendency of variation for different samples is not convincing. It is better to add one sample with the annealing temperature around 663 K.

<Reply>: We have observed that the largest S_{ANE} is obtained when the number density of Cu nano-clusters is maximized in an amorphous matrix. This occurs only prior to Fe crystallization starting at ~660 K. Increasing the temperature above 653 K, i.e., 663 K, results in formation of α -Fe nanocrystals and it is detrimental for S_{ANE} . In fact, the window in which we maximize the number density of Cu nanoclusters is rather narrow. Nevertheless, even considering the error bar in S_{ANE} in Fig. 4, 653 K annealed sample shows the largest value of S_{ANE} .

Response to Reviewer 3

<Reviewer 3>: The paper by R. Gautam et al. presents an interesting improvement of the anomalous Nernst effect (ANE) in flexible Fe-based amorphous materials through nonstructural engineering. However, because of the presence of some deficient in the results, we suggested that the manuscript in the present form is not suitable for publication in the journal of Nature Communications. The reason is as follows: Though the authors give a analysis of the evolution of the Cu nano-clustering with annealing temperature by TEM and APT, to understand the improvement of the ANE in detail, the particle size of the Cu clustering and its content in the composite should be quantitatively estimated.

<Reply>: We analyzed the average size and number density of Cu clusters using APT data. The results are presented in Supplementary Fig. S4, which illustrates the variation of cluster size and number density with annealing temperature. We have also included these results in the manuscript in page 7 as follow:

“As the annealing temperature was raised to 653 K, Cu clustering became more visible as demonstrated in Fig. 3c, with an increase in both the size and number density of nano-sized Cu-rich clusters (Supplementary Fig. 4). The average size of the Cu clusters for the Nanomet sample annealed at 653 K was determined to be approximately 2.2 nm, with an estimated number density of $7.3 \times 10^{23} \text{ m}^{-3}$.”

Fig. S4 | Variation in the number density and size of Cu clusters for Nanomet samples as a function of annealing temperature. The white region indicates the presence of Cu-clusters in an amorphous matrix, while the grey dotted line shows the onset of primary crystallization and grey regions signify the presence of α -Fe crystalline phase and Cu-clusters in an amorphous matrix.

The red dotted line indicates the annealing temperature (653 K) that exhibits the highest SANE value.

<Reviewer 3>: To estimate the contribution of Cu nano-clustering in the thermal electric effect in the Fe based Nanomet, the thermal conductivity and ANE effect of the pure Cu particle are required.

<Reply>: We appreciate the reviewer for this comment. Thermal conductivity of pure Cu is known to be $385 \text{ WK}^{-1}\text{m}^{-1}$ according to literature [Hust, J. G. THERMAL CONDUCTIVITY OF Al, Cu, Fe AND W. CODATA Bull. **59**, 29–45 (1985)]. However, ANE is realized in the magnetic composite materials. Thermoelectric performance should be evaluated by effective values for the composite materials; the high thermal conductivity of Cu may increase the total thermal conductivity of the composite materials, but the thermal conductivity value of Cu itself does not directly determine the thermoelectric performance. We have measured the thermopower and thermal conductivity as effective values that include the effect of Cu clusters, and an argument based on the parameters obtained by our measurements is appropriate.

<Reviewer 3>: To improve the level of the manuscript, a theoretical analysis of the improvement of the ANE effect with the nonstructural engineering is highly desired.

<Reply>: We thank the reviewer for the constructive suggestion. We agree that understanding the underlying physics of the observed increase in S_{ANE} realized by nanostructure engineering is very important. The amorphous nature and complicated microstructure of our materials impose various limitations on the theoretical analysis. However, to shed a light on the origin of an increase in S_{ANE} by nanostructure engineering, we conducted the anomalous Hall and Seebeck effect studies to understand the origin of ANE by estimating the transverse thermoelectric conductivity. The corresponding results and discussion are included in the revised manuscript on pages 10 and 11 as follows:

“ S_{ANE} can be described as two components (S_{I} and S_{II}) using the linear response equation¹⁷: $S_{\text{ANE}} = \rho_{xx}\alpha_{xy} - \rho_{\text{AHE}}\alpha_{xx} \equiv S_{\text{I}} + S_{\text{II}}$. Here, ρ_{xx} , ρ_{AHE} , α_{xy} and α_{xx} represent the longitudinal electrical resistivity, anomalous Hall resistivity, transverse thermoelectric conductivity, and longitudinal thermoelectric conductivity, respectively. The intrinsic ANE that directly converts temperature gradient into a transverse voltage is signified by the component S_{I} , whereas the S_{II} reflects the voltage originating from the Hall effect of the longitudinal charge flow induced by the Seebeck effect (SE) due to the anomalous Hall effect (AHE). The S_{II} can be rewritten as $\frac{-S_{xx}\rho_{\text{AHE}}}{\rho_{xx}} = -S_{xx}\tan\theta_{\text{AHE}}$, where S_{xx} is the Seebeck coefficient and θ_{AHE} is the anomalous Hall angle. To discern the S_{I} and S_{II} contribution to ANE, we estimated the θ_{AHE} and α_{xy} by conducting transverse electric resistivity (ρ_{yx}) and S_{xx} measurements at room temperature for the Nanomet samples annealed at 643 K and 653 K. This enables us to gain better understanding on the origin of an increase in S_{ANE} . Table 1 illustrated the estimated values, revealing that the magnitude of S_{II} in both samples is considerably smaller compared to S_{I} . This infers that ANE is primarily governed by the S_{I} components due to the large value of α_{xy} , while the impact of the anomalous Hall and Seebeck effects is negligible. The α_{xy} value for the Nanomet samples annealed at 643 K and 653

K was 1.41 and 1.93 $\text{Am}^{-1}\text{K}^{-1}$, respectively. This significant difference in α_{xy} values is consistent with the trend observed in S_{ANE} and can be attributed to the presence of a high density of nano-size Cu clusters in amorphous matrix.”

REVIEWER COMMENTS

Reviewer #1 (Remarks to the Author):

I had minor questions to the authors, which were well treated in the revised manuscript. I like to recommend this version to be accepted as is.

Reviewer #2 (Remarks to the Author):

The manuscript has been well revised based on the comments and suggestions from the referees. I would like to conclude that the current manuscript is suitable for publication in NC.

Reviewer #3 (Remarks to the Author):

In the revised manuscript, the authors essentially provided a fair answer to the comments of the reviewers. However, it is also noted that the author mentioned in line 253 of the revised article that the generation of the giant ANE “can be attributed to the presence of a density of nano-size Cu clusters in amorphous matrix”. However, further experiments are needed to confirm this conclusion. Firstly, it seems that the results of the article were based on the only one sample with nanostructure engineering. In order to increase the credibility of the data, the author needs to test on different samples with the nanostructure engineering. Secondly, in order to verify that the conclusion that “the higher the copper particle density, the higher the ANE”, can the authors increase the copper content to prepare samples with higher copper density?

Response Letter

We appreciate the reviewers for reading our manuscript and giving us constructive comments. We incorporated all the comments in the revised version, which are indicated in red in the submitted manuscript. How we revised the manuscript based on the comments are listed below:

Response to Reviewer 3

<Reviewer 3>: In the revised manuscript, the authors essentially provided a fair answer to the comments of the reviewers. However, it is also noted that the author mentioned in line 253 of the revised article that the generation of the giant ANE “can be attributed to the presence of a density of nano-size Cu clusters in amorphous matrix”. However, further experiments are needed to confirm this conclusion. Firstly, it seems that the results of the article were based on the only one sample with nanostructure engineering. In order to increase the credibility of the data, the author needs to test on different samples with the nanostructure engineering.

<Reply>: We appreciate the reviewer for the insightful comment. Indeed, our results have been validated across several samples exhibiting different nanostructure features but with the same composition. To provide more compelling evidence on the effect of Cu cluster number density and their influence on ANE, we conducted an additional atom probe tomography (APT) analysis on a different nanostructure-engineered sample (annealed at 643 K, S_{ANE} value of 2.74 $\mu\text{V/K}$). Supplementary Fig. 4 illustrates the APT elemental maps and shows the variation of cluster size and number density with annealing temperature.

Although our paper primarily emphasizes nanostructure engineering rather than enhancing the ANE coefficient through alloy composition modification, to further support our results, we have prepared another Fe-based amorphous material ($\text{Fe}_{80.5}\text{Si}_{13.8}\text{B}_{13.6}\text{Cu}_{1.5}\text{C}_{0.6}$ in at.%) with a higher Cu content and evaluated its transverse thermoelectric conversion properties. The new results are provided in Supplementary Figs. 12-15. Following our initial findings, wherein the maximum Cu nanoclusters were observed in proximity to the onset temperature of crystallization, we conducted differential scanning calorimetry (DSC) on as-quenched $\text{Fe}_{80.5}\text{Si}_{13.8}\text{B}_{13.6}\text{Cu}_{1.5}\text{C}_{0.6}$ ribbons as demonstrated in Supplementary Fig. 12a and conducted annealing at different temperatures as X-ray diffraction (XRD) results are shown in Supplementary Fig. 12b. Due to the large Cu content of the alloy composition, the as-quenched sample revealed Cu clusters in an amorphous matrix, in contrast to the $\text{Fe}_{84.7}\text{Si}_{2.8}\text{B}_{3.8}\text{P}_{7.8}\text{Cu}_{0.7}\text{C}_{0.2}$ composition ribbons presented in the main text of the manuscript. However, upon annealing at 646 K, we observed a fourfold increase in Cu cluster density (Supplementary Fig. 13). Consequently, we achieved an enhancement in the S_{ANE} from 2.63 μVK^{-1} to 3.48 μVK^{-1} as demonstrated in Supplementary Fig. 14. These results provide additional support that nanostructure engineering in ribbons with different compositions can indeed lead to an increase in ANE. However, composition modification is not the primary focus of our paper. Nonetheless, these results serve as supplementary evidence that ANE can indeed be enhanced by nanostructure engineering.

The corresponding results and discussion are included in the revised manuscript on page 13 and in Supplementary Information as a new section as follows:

In the manuscript, page 13

“Furthermore, we validated our nanostructure engineering approach in tailoring ANE using another Fe-based amorphous material with a higher Cu content (see Supplementary Figs. 12-15). Increase of Cu content in the alloy composition to 1.5 at.% led to an increase in the PF to $\sim 8.9 \mu\text{Wm}^{-1}\text{K}^{-2}$ and $Z_{\text{ANE}}T$ to $\sim 2.5 \times 10^{-4}$ (see Supplementary Fig. 15).”

In Supplementary Information, pages 2, 5 and 12-15

“Nanostructure engineering of Fe-based amorphous materials with high Cu content

To validate the efficacy of our nanostructure engineering approach in tailoring the ANE, we prepared another Fe-based amorphous material with a higher Cu content using the melt-spinning technique. The composition of this material was estimated using inductively coupled plasma optical emission spectrometry (ICP-OES) and was found to be $\text{Fe}_{80.5}\text{Si}_{3.8}\text{B}_{13.6}\text{Cu}_{1.5}\text{C}_{0.6}$ (at.%). DSC analysis of the as-quenched sample reveals the crystallization temperature of the α -Fe(Si) phase at 656 K, as depicted in Supplementary Fig. 12a. The as-quenched ribbons were annealed at various temperatures (626 K, 646 K, and 656 K) to facilitate the formation of high-density nano-sized Cu-clusters. X-ray diffraction (XRD) analysis, presented in Supplementary Fig. 12b, indicates the presence of an amorphous phase for samples annealed below 646 K, while the sample annealed at 656 K exhibited the formation of α -Fe(Si) crystalline phase. Atom probe tomography (APT) analysis of both as-quenched and 646 K annealed samples was carried out to study the formation of Cu-clusters, as illustrated in Supplementary Fig. 13. Due to the higher Cu content, APT elemental mapping of the as-quenched sample reveals the presence of Cu-clusters in an amorphous matrix. Annealing at 646 K led to a fourfold increase in the number density of Cu-clusters. Notably, the increase in Cu-cluster density also resulted in the enhancement of the S_{ANE} value from $2.63 \mu\text{VK}^{-1}$ to $3.48 \mu\text{VK}^{-1}$, as demonstrated in Supplementary Fig. 14. These findings provide additional evidence that nanostructure engineering in ribbons with different compositions can indeed result in an increase in ANE.

An increase in Cu-content from $\text{Fe}_{84.7}\text{Si}_{2.8}\text{B}_{3.8}\text{P}_{7.8}\text{Cu}_{0.7}\text{C}_{0.2}$ to $\text{Fe}_{80.5}\text{Si}_{3.8}\text{B}_{13.6}\text{Cu}_{1.5}\text{C}_{0.6}$ led to an increase in Cu cluster density from $7.3 \times 10^{23} \text{ m}^{-3}$ to $9.05 \times 10^{23} \text{ m}^{-3}$, respectively. Importantly, the high Cu content Fe-based amorphous material holds merits in terms of power factor and figure of merit values due to enhanced electrical conductivity, as shown in Supplementary Fig. 15. However, the high Cu content sample shows a smaller S_{ANE} value, which can be attributed to alterations in the composition of the matrix amorphous phase. Therefore, the ANE can be further amplified by employing compositional as well as nanostructure engineering.”

Supplementary Fig. 4 | APT analysis of Nanomet sample annealed at 643 K and variation in the number density and size of Cu clusters for Nanomet samples as a function of annealing temperature. a, APT elemental maps of Fe (blue), P (red), B (green), and Cu (orange) for Nanomet sample annealed at 643 K. **b,** Annealing temperature dependence of the Cu-cluster number density and average Cu-cluster size. The white region indicates the presence of Cu-clusters in an amorphous matrix, with the grey dotted line denoting the onset of primary crystallization and grey regions signify the presence of α -Fe crystalline phase and Cu-clusters in an amorphous matrix. The light gold region represents the annealing temperature associated with a high-density region of Cu-clusters.

Supplementary Fig. 12 | DSC and XRD analysis of high Cu content Fe-based amorphous material. a, DSC curve of the as-quenched melt-spun ribbon and **b**, XRD pattern of high Cu content Fe-based amorphous samples ($\text{Fe}_{80.5}\text{Si}_{3.8}\text{B}_{13.6}\text{Cu}_{1.5}\text{C}_{0.6}$ at.%) annealed at 646 K and different time.

Supplementary Fig. 13 | APT analysis of high Cu content Fe-based amorphous material. APT elemental maps depict the distribution of Fe (blue), Si (grey), B (green), and Cu (orange) in the as-quenched sample (**a**) and sample annealed at 646 K (**b**). **c**, Estimated cluster size and number density of both as-quenched and annealed samples.

Supplementary Fig. 14 | Transport properties of high Cu content Fe-based amorphous samples. a-d, Annealing temperature dependence of the thermal diffusivity D and specific heat C_p (a), thermal conductivity κ and electrical conductivity σ (b), anomalous Nernst coefficient S_{ANE} and the corresponding anomalous Ettingshausen coefficient Π_{AEE} estimated using the Onsager reciprocal relation at 300 K (c), and power factor (PF) and dimensionless figure of merit for ANE $Z_{\text{ANE}}T$ at 300 K (d) for the high Cu content Fe-based amorphous samples.

<Reviewer 3>: Secondly, in order to verify that the conclusion that “the higher the copper particle density, the higher the ANE”, can the authors increase the copper content to prepare samples with higher copper density?

<Reply>: We appreciate the reviewer's suggestion and have prepared new samples with a higher copper content. An increase in Cu content may impact the formation of amorphous + α -Fe(Si) crystals in the as-melt-spun state. However, through careful optimization of the alloy composition to Fe_{80.5}Si_{13.8}B_{13.6}Cu_{1.5}C_{0.6} (at%), we have successfully achieved Fe-based amorphous ribbons with a higher Cu content. As mentioned by the reviewer, an increase in Cu content leads to a higher density of Cu cluster; i.e. $7.3 \times 10^{23} \text{ m}^{-3}$ for Cu0.7 at.% sample to $9.05 \times 10^{23} \text{ m}^{-3}$ for Cu1.5at.% sample. Importantly, we observed that the high Cu content Fe-based amorphous material has merits in the power factor and figure of merit values due to enhanced electrical conductivity, as shown in Supplementary Fig. 15. However, due to the change in the composition of the matrix amorphous phase, comparing S_{ANE} cannot be solely attributed to the Cu cluster density. Therefore, two factors of composition modification and Cu cluster density are playing a role in determining S_{ANE} . Hence, a thorough investigation is required to optimize the composition as well as nanostructure engineering of Cu clusters to further increase ANE, which is beyond the scope of the current paper with a primary focus on nanostructure engineering.

Supplementary Fig. 15 | Comparison of properties between low and high Cu content Fe-based amorphous samples. a-f, Thermal diffusivity D (a), thermal conductivity κ (b), electrical conductivity σ (c), anomalous Nernst coefficient S_{ANE} (d), Power factor (PF) (e), and dimensionless figure of merit for ANE $Z_{\text{ANE}} T$ at 300 K (f) between low and high Cu content Fe-based amorphous samples. Samples with the highest ANE values are chosen for this comparison.

REVIEWERS' COMMENTS

Reviewer #3 (Remarks to the Author):

Dear editor.

The manuscript has been well readvised based on the comments from the referee. I like to recommend this revised version to be accepted for publication in NC.

Additionally, more references about the ANE (such as. Nat. Mater. 2021, 20, 463 (2021); Adv. Mater. 35, 2301339(2023)) are suggested, without compulsion, to be cited in the introduction part, which may helps to extend the understanding of the background of this study.

Thanks.

Response Letter

We appreciate the reviewers for reading our manuscript one more time. The final request of the reviewer is also considered in our manuscript as follow:

Response to Reviewer 3

<Reviewer 3>: The manuscript has been well readvised based on the comments from the referee. I like to recommend this revised version to be accepted for publication in NC. Additionally, more references about the ANE (such as. Nat. Mater. 2021, 20, 463 (2021); Adv. Mater. 35, 2301339(2023)) are suggested, without compulsion, to be cited in the introduction part, which may helps to extend the understanding of the background of this study.
Thanks.

<Reply>: We appreciate the reviewer for the positive comments and accepting our manuscript as it is. These two references are already cited in the revised manuscript (reference numbers of 27 and 48).